# Solution: Inspecting Traffic in Residential Networks with Opportunistically Outsourced Middleboxes

## Abstract

Since they lack the powerful tools and personnel available in enterprise-grade security solution, home networks have particularly difficult network security challenges. While prior efforts outsource network traffic to cloud or cloudlet services, such measures redirect network traffic out of the home network, which grants a third-party access to see and profile traffic. This affects the privacy of that traffic. Further, if those tools need to apply Transport Layer Security (TLS) decryption to enhance their monitoring insight, the privacy risks to home users grows substantially. Alternatively, residents may introduce new physical hardware in their home networks, but doing so incurs greater capital costs that would impede deployment.

Our work explores a system to leverage existing available devices, such as smartphones, tablets and laptops, inside a home network to create a platform for traffic inspection. By using devices owned and operated by the same end-users, the system can peeking into TLS traffic and perform detailed inspection without introducing risks from third parties. By leveraging existing devices in a home network, we can implement our platform with no additional hardware costs. Our performance evaluation shows that such middleboxes can substantially increase the throughput of communication from around 10 Mbps to around 90 Mbps, while increasing CPU usage at the router by around 15% , with a 20% CPU usage increase on a smartphone (with single core processing), and with a latency increase of about 120 milliseconds to network packets.

## 1 Introduction

The increasing use of household broadband Internet service and smart home technologies are likewise increasing risks related to home network. In 2021, 442 million smart devices, and 82% home networks are connected to the Internet. New smart devices, like smart cameras, can collect data from users to provide intelligent services, but it can be difficult for defenders to determine what network traffic is associated with these devices. It can be difficult for users to determine if their devices are safe from network attacks. A study with fifteen smart home users indicates that eleven participants were worry about physical security and five participants were concerned about the privacy associated with these devices [1].

Typically, consumer-grade routers do not provide effective network protections [2]. As a result, attackers have various opportunities to compromise networked devices, allowing lateral propagation and a range of attacks.

The limited capabilities of consumer-grade network hardware force difficult trade-offs in modern home networks. While prior work has proposed lightweight functionality on residential routers [3], the computational constraints of those devices limit the types of tasks that can be hosted on such routers. Efforts to profile and examine encrypted traffic using machine learning [4] would exceed the resources of many such routers. These routers are unable to engage in the more sophisticated analysis common in enterprise security gateways.

Other techniques push the computational tasks associated with network screening to remote servers. Feamster [5] proposes that home networks can outsource their security mechanisms to cloud servers with software-defined networking (SDN) technologies. TLSDeputy [6] uses remote servers to validate TLS certificates and protocol settings to ensure the authenticity of communicating endpoints. However, both techniques allow the operators of cloud infrastructure to have insight into the activities of a home network, introducing new privacy risks and an expanded trusting computing base.

In contrast to prior efforts, we consider mechanisms to deploy home network traffic inspection in an opportunistic fashion. We explore mechanisms to leverage existing devices in a home network when they are available to screen communication. In doing so, we ask the following research questions:

- To what extent can we utilize current resources within a home network to build real-time packet inspection?

- To what extent would such a packet inspection system influence the performance of the home network, in terms of traffic latency, resource consumption, and throughput?

Our approach enables devices such as smartphones, tablets, laptops and desktops to perform traffic analysis. These devices can operate as security proxies when they are available, enabling detailed analysis. In pursuing this direction, our work makes the following contributions:

- **Creation of Prototype On-Router and Outsourced Middleboxes:** We use open source firmware on a consumer-grade router to profile traffic locally and via a

smartphone. We compare baseline communication with an on-router program that profiles destination addresses. We further implement a technique to transparently direct traffic through a smartphone middlebox using network address translation (NAT) rules on the router.

- **Performance Evaluation of Deployment Options:** We compare baseline forwarding of the router with on-router inspection and with opportunistic outsourcing to a smartphone. Our evaluation shows that on-device inspection has a throughput of around 10 Mbps whereas outsourcing the inspection to a smartphone achieve roughly 90 Mbps of throughput. The smartphone middleboxing approach adds around 15% CPU usage to the router, 20% CPU usage to a smartphone (with single core processing), and introduces 120 milliseconds of round trip time (RTT) delay to network traffic.

## 2 Background and Related work

In this section we introduce the background knowledge and prior work on residential network computation and security.

### 2.1 Computation in Residential Networks

A 2015 survey found that 77% of US households subscribed broadband internet service, and 78% own a desktop or laptop computer [7]. However, modern home networks face many security challenges, since the value of assets managed by residential networks is increasing. Attackers can gain sensitive information or directly control the devices and launch attack on other devices, such eavesdropping, replay attack, network scanning, and data theft [8, 9].

There are effective ways to detect these attacks, but they require significant computational resources. Hafeez *et al.* [8] find that machine learning methods can detect a series attack with accuracy as high as 99%. Jan *et al.* [10] propose a method to detect a compromised device that joins a botnet with very limited data through a deep learning algorithm. A powerful inspection platform is helpful in increasing home network security.

### 2.2 Perimeter Defense for Home Networks

Perimeter defenses can be useful for residential networks. While the basic NAT functionality on residential routers typically prevents unsolicited inbound communication, it is ineffective at detecting or stopping existing compromises within a network or attacks that are launched via a connection initiated from within the network.

Li *et al.* propose applying deep learning anomaly detection techniques for securing home networks; however, their method runs on equipment with computational resources that may be inapplicable to many home networks [11].

ParaDrop [12] proposes allowing third-party application providers to install lightweight containers to provide a gateway for simple tasks. However, ParaDrop does not have sufficient resources to run resource-consuming tasks like intrusion detection. Another work [13] adds plug-and-play devices to a consumer-grade router, which enables the router to work as an intelligent IoT gateway that can inspect traffic; however, it incurs capital costs and requires hardware modifications inside consumer routers that are likely beyond the technical abilities of some potential deployers.

Shirali-Shahreza *et al.* [14] summarized commercial home network firewall products. Each requires the installation of additional devices in the network with an initial cost of at least $200 and with ongoing monthly service costs. These devices may replace typical home routers or act in conjunction with existing routers. Some use virtual private network (VPN) techniques to tunnel traffic to a remote VPN server that can inspects and analyze home network traffic before forwarding the traffic. These methods introduce additional costs and equipment for users.

To simplify the management but keep security enforcement, Feamster proposes to outsource security needs to a remote cloud server through SDN architecture [5]. Experts and professional security software help to dynamically manage the network. Since cloud servers may contain richer resources, modules running in the controller can provide better analysis, along with a broader view of the network. Many other works propose to build remote firewalls on the controller based on this architecture [8, 14–18]. Most of them propose to utilize the gateway as an OpenFlow host in the home network. Others propose to use a locally-available device, such as a Raspberry Pi, instead. The agent usually samples the network traffic and uploads it to the cloud. Controllers running in the cloud may also run a firewall module to inspect the sampled flow information. The agent further executes the returned action decision, which is usually a security policy received from the controller. However, these methods are based on users' trust that their private data is being used properly by a third-party provider; some users have concerns about such providers.

### 2.3 Edge Computing for Local Networks

The edge computing paradigm builds decentralized computing pools for processing jobs from clients, bringing the computation closer to the source of data [19]. Cloudlet [20] is a popular edge computing prototype that offloads tasks to nodes that can scale. These nodes can be hosted by ISPs or other providers. Drop computing [21] builds a collaborative computing cloud using mobile devices in which one device can offload tasks to other devices. When there is no available device, the system seeks help from the cloud. This method is designed for ad hoc networks, which lack reliability since devices may enter and leave the network frequently and network coverage is usually limited. Similarly, Verbelen *et al.* [22]

split tasks and offload them to a virtualized environment, either on mobile devices or on the cloud. Gedeon *et al.* propose to let a more reliable device, a home network gateway, run a broker. The gateway seeks available cloudlet nodes to help with its tasks [23]. Their methods still outsource the computation to a third-party platform, which raises privacy concerns. That work demonstrates that running a broker on a residential router does not introduce significant overhead, a result we leverage in our own approach.

Aazam *et al.* [24] propose to use either smart gateway or other localized fog nodes to do data pre-processing, before uploading data to the cloud. The pre-processing not only reduces the data size and retains only the necessary data, it can reduce some, but not all, privacy risks.

We explore a mechanism to perform network traffic inspection within a home network only. Unlike existing edge computing work, we focus on inspecting streaming data rather than discrete computational jobs.

# 3 Approach

Our research compares two approaches: on-router inspection via `NFQUEUE` and on-phone inspection via NAT redirection. We start by introducing the threat model. Then, we illustrate the process of on-router inspection. Finally, we describe the functionalities of each component of the phone-based inspection platform and how they work together.

## 3.1 Threat Model and Scope

We consider a basic threat model where malicious communication occurs between a host within the network and a system outside of the home network. The defender's goal is to inspect all traffic leaving the home network. In this model, we trust the router and the smartphone that acts as a proxy. We do not trust the endpoints.

Our goal is to create mechanisms that enable arbitrary traffic inspection on a reasonably resourced device, such as a phone. We do not aim to create new anomaly detectors or traffic inspection engines; that is out of the scope of this work. Accordingly, we demonstrate baseline functionality using an block list of destination addresses.

## 3.2 On-Router Inspection via `NFQUEUE`

We implement a basic C++ program to inspect IP addresses that is compiled to run natively on the router. The program uses the `iptables` packet inspection tool and the `netfilter_queue` library to inspect traffic. Essentially, the `iptables` tool operates on each packet processed by the Linux stack on the router. This action occurs when packets cross from the LAN interfaces to the WAN interface. The `iptables` program sets an `NFQUEUE` judgment for all packets,

causing them to enter a kernel data structure. The C++ program extracts the packets from that data structure, inspects the address, and returns the packets to the kernel queue for transmission. This program represents the minimum inspection required for a general-purpose user-space inspection program on the router.

## 3.3 On-Phone Inspection via NAT Redirection

There are two components in our on-phone inspection approach. The first is a set of NAT rules on the router. We use the `iptables` program, which can manage IP packet rules in the Linux kernel. The NAT table is one table of `iptables` to create several rules in the NAT table to transform the original destination IP address of the packets from the server to the IP address of the smartphone, so the traffic sent from the client can be redirected to the smartphone. In the example shown in Figure 1, we first apply a DNAT rule as `iptables -t nat -A PREROUTING -p tcp -s 192.168.1.2 -d 172.16.1.2 -dport 6666 -j DNAT -to-destination 192.168.1.3:6666` and an SNAT rule as `iptables -t nat -A POSTROUTING -p tcp -s 192.168.1.2 -d 192.168.1.193 -dport 6666 -j SNAT -to-source 192.168.1.1` to forward traffic to the smartphone. Then the smartphone works as a proxy that receives packets and sends them back to the router after inspection. When these packets return to the router, the router transforms their destination IP address to the original server destination IP address based on another DNAT rule, such as `iptables -t nat -A PREROUTING -p tcp -s 192.168.1.3 -d 192.168.1.1 -sport 7777 -j DNAT -to-destination 172.16.57.216:6666`. Since all of the NAT rules work bidirectionally, the packets sent from server will will also go in the reverse direction, again traversing the smartphone. Rather than processing traffic as an arbitrary user space program in the router's Linux stack, our method forwards them using kernel data structures. This feature avoids potentially costly transitions to user space.

The second component in our approach is the proxy device and service. We first explore the smartphone as a proxy and implement a Java program that uses TCP to accept traffic for inspection on a pre-defined port. Figure 1 shows how the phone accepts communication from the router on a specific port. It starts a new TCP connection to a specifically configured port on the router, which the router pre-configures to forward to the remote server. Since the smartphone is on-path in our method, we retrieve the raw payload of every packet. While we only apply IP list filtering, more advanced inspection can be deployed in our method, such as TLS inspection (TLSI) .

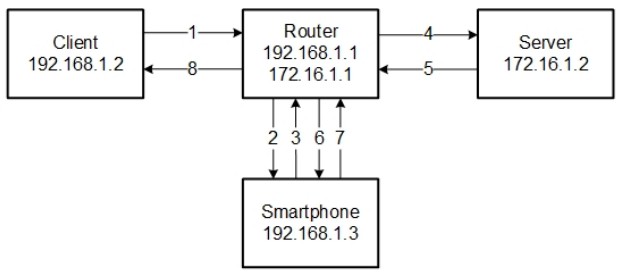

| Packet # | Source IP | Source Port | Destination IP | Destination Port |
|---|---|---|---|---|
| 1 | 192.168.1.2 | 47289 | 172.16.1.2 | 6666 |
| 2 | 192.168.1.1 | 6001 | 192.168.1.3 | 6666 |
| 3 | 192.168.1.3 | 7777 | 192.168.1.1 | 6001 |
| 4 | 172.16.1.1 | 6002 | 172.16.1.2 | 6666 |
| 5 | 172.16.1.2 | 6666 | 172.16.1.1 | 6002 |
| 6 | 192.168.1.1 | 6001 | 192.168.1.3 | 7777 |
| 7 | 192.168.1.3 | 6666 | 192.168.1.1 | 6001 |
| 8 | 172.16.1.2 | 6666 | 192.168.1.2 | 47289 |

Figure 1: An example of packet forwarding via NAT rules. As the client sends the original packet to the server, the router modifies the packet and forwards it to the smartphone. After the smartphone performs packet inspection, it sends the packet back to the router. Then the router forwards it to the server. Since all of the NAT rules work bidirectionally, the packets sent from the server will follow the reverse path.

## 4 Implementation

We implement our method in a lab environment on physical devices. We run the OpenWrt 21.02.2 operating system (OS) on a TP-LINK AC1750 Wireless Dual Band Gigabit Router. We simulate a home network client user on a laptop with four cores and 16 GBytes of memory, running the Windows OS. We simulate a server outside of the home network on a laptop with four cores and 16 GBytes of memory, running the Ubuntu 20.04 OS. We use a smartphone with eight 2.0 GHz cores and 4 GBytes of memory, running the Android 11 OS as the proxy device.

For the network configuration, as shown in Figure 2, we create two VLANs: one is on interface `eth0`, and the other is on interface `eth1`. We assign the LAN ports and wireless radio to one VLAN, and assign the WAN port to the other VLAN. The client connects to a LAN port via a category 6 Ethernet cable. The server also connects to the WAN port using a category 6 cable. For the radio, we build an access point on 5.785 GHz using a Qualcomm Atheros QCA 9880 802.11ac adapter. We connect the smartphone to this access point at a distance of 3 feet with an unobstructed line of sight.

After configuring the home network as defined in the threat model, we add three NAT rules to `iptables` in the router, as described in Section 3. These rules include SNAT and DNAT rules and have the capability of redirecting traffic between the client and the server to travel via the smartphone. On the smartphone side, we use Android Studio to build a

Java application that performs packet inspection based on a malicious IP block list and hosts a proxy service.

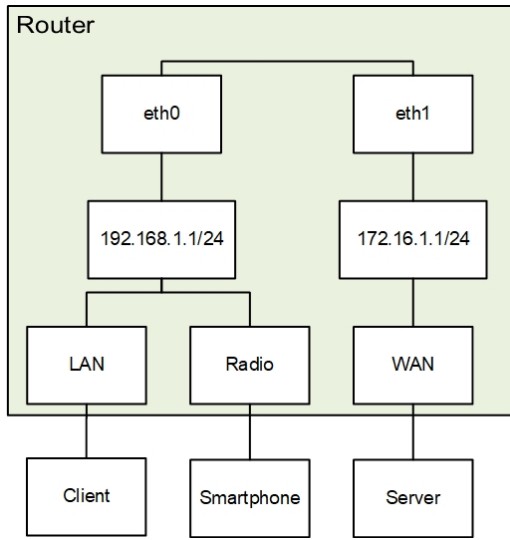

Figure 2: Our implementation's network configuration

## 5 Performance Evaluation and Results

The most straightforward mechanism for implementing an inspection and analysis middlebox is to use a device that is already physically on the network path. In home networks, a residential wireless router typically fills that role. To justify the added complexity of opportunistic middleboxes, we explore the performance implications of using such commodity devices. We use a typical network setting, without the use of inspection functionality, to establish a baseline. We then explore on-device inspection. Finally, we examine an inspection method in which NAT rules are used to reroute traffic to a middlebox, using both a smartphone emulator and a physical commodity smartphone for analysis.

In examining these scenarios, we evaluate the performance of each using four metrics: flow throughput, end-to-end round trip time (RTT), the CPU usage at the router, and the CPU usage of the smartphone when it is in use.

### 5.1 The Baseline: LAN to WAN traffic

Our baseline scenario connects a client to a server though a residential router. Often, the WAN port is used on the router to connect to upstream networks, such as the Internet, and the servers available through those networks. Therefore, we connect an Ubuntu server to the WAN port of the router using a category 6 Ethernet cable, which supports duplex gigabit connectivity. The server uses a gigabit Ethernet card. We statically configure the IP addresses of the server and the

router's WAN port within a subnet that is only used by those two devices.

The connectivity options for clients may vary in different homes. Some devices may be connected via Ethernet connections to the LAN ports of the router. In other cases, devices may connect using WiFi radio links. Accordingly, we explore both of these connection scenarios.

We begin by exploring the case in which the client is connected to a LAN port on the router via a category 6 Ethernet cable. We use the router's built-in DHCP server, which assigns an address to the client in a subnet that the router and client share, yet is disjoint from the subnet used by the server. We use the router's built-in default NAT capabilities to translate across the subnets, which is a common deployment model in homes. Using the `iperf3` benchmarking tool [25], we test a TCP connection between the client and the server. We configure `iperf3` to attempt to maximize throughput in the channel and observe it for 1,100 seconds. We conducted 3 trials and measured the throughput for 1000 seconds after an initial delay of 100 seconds to accommodate TCP's slow-start behavior. As we see in the right-most two lines in Figure 3, the median download throughput is 440.00 Mbps and the median upload throughput is 254.00 Mbps, with tight distributions (standard deviation of 4.90 Mbps for download and 3.27 Mbps for upload).

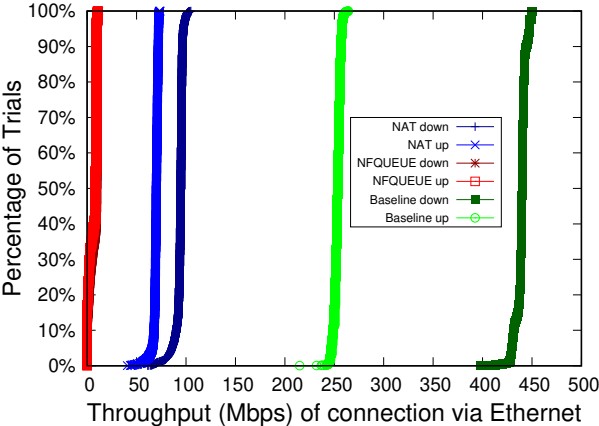

Figure 3: Results from throughput tests when the client connects to the router via a category 6 Ethernet cable. The green lines show upload and download throughput under a baseline setting. The red lines show both throughputs after applying on-router inspection via `NFQUEUE` library. The blue lines show both throughput after applying on-phone inspection using NAT redirection rules.

Next, we determine the impact of connecting the client to the router using WiFi radios. The client has a network adapter capable of using 802.11ac communication, and the router supports 802.11ac, leading to the lowest common denominator

of the 802.11ac standard. That standard has a theoretical maximum throughput of 1300 Mbps, though practical throughput is often less due to interference and obstructions. We place the router and the client roughly 3 feet apart with line-of-sight. We then repeat our throughput analysis using the `iperf3` benchmark tool, with the same settings as our Ethernet experiments. We use the same 3 trials and timing windows as in the earlier experiment. In Figure 4, the right-most two lines show the baseline throughput results via WiFi. The median download throughput is 196.00 Mbps and the median upload throughput is 229.00 Mbps, with standard deviation of 10.38 Mbps for download and 14.47 Mbps for upload.

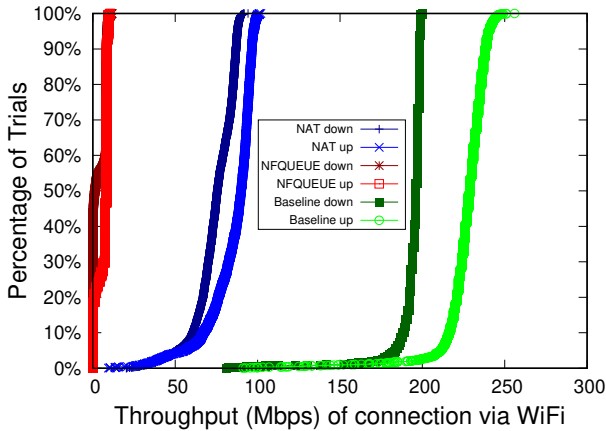

Figure 4: Results from throughput tests when the client connects to the router wireless. The rightmost green lines show upload and download throughput under a baseline setting. The leftmost red lines show the throughput after applying on-router inspection via `NFQUEUE` library. The middle blue lines show the throughput after applying on-phone inspection using NAT redirection rules.

Since the communication throughput via Ethernet appears to be less than the medium's theoretical maximum, we explore whether the router could be causing a bottleneck. In particular, we examine the CPU of the router. While we test the maximum throughput, we use the `top` tool to record the CPU usage of the router for 1000 seconds. As shown in Table 1, the CPU usage of the router is at its limit more than 90% of the time when testing maximum throughput. These results show that the CPU of our router is the performance bottleneck for higher throughput.

To determine the added CPU usage from different traffic inspection methods, we need to measure the router's CPU usage in a moderate working scenario, rather than in an extreme situation. We thus evaluate the scenario in which the TCP connection throughput is reduced to 10 Mbps of randomized payload to the server. We also record the CPU usage of the router for 1000 seconds. The green line in Figure 5 shows that

Table 1: CPU usage of the router while testing the maximum throughput of connections in six scenarios.

| Percentile of Trials | 10th | 50th | 90th |
|---|---|---|---|
| CPU Usage in Baseline Upload | 100% | 100% | 100% |
| CPU Usage in Baseline Download | 100% | 100% | 100% |
| CPU Usage in NAT Upload | 98% | 100% | 100% |
| CPU Usage in NAT Download | 97% | 100% | 100% |
| CPU Usage in NFQUEUE Upload | 100% | 100% | 100% |
| CPU Usage in NFQUEUE Download | 100% | 100% | 100% |

the median CPU usage of the router is 9.00% with standard deviation of 1.58%.

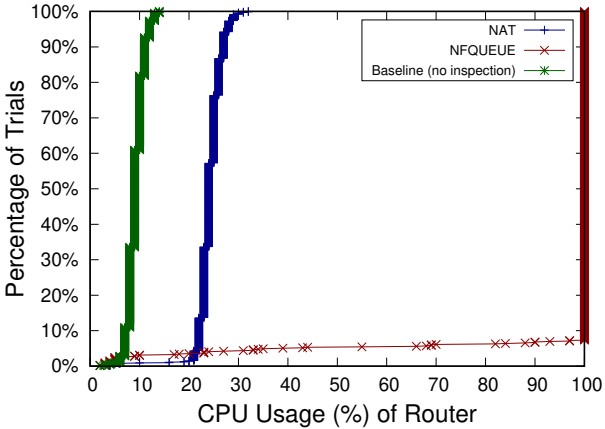

Figure 5: CPU usage of the router when applying on-phone inspection using NAT redirection rules, applying on-router inspection via NFQUEUE library, and a baseline without inspection when throughput is limited to 10 Mbps.

While throughput is an important metric, the end-to-end round trip time (RTT) is also important for understanding the delay introduced by the network paths and the router. To test this, we construct an echo program on the server and a recording device on the client to measure the time difference between the client sending a specific payload and receiving a reply. Across 1000 trials, we see that the left-most line in Figure 6 has a median RTT of 1.12 ms with a standard deviation of 0.12 ms. When repeating this analysis via WiFi, in Figure 7, we see the median RTT of the left-most line is 2.72 ms with a 6.14 ms standard deviation.

## 5.2 On-Router Inspection via NFQUEUE

To explore whether the router itself can feasibly inspect traffic, we implement a basic C++ program, that is compiled to run natively on the router, to inspect IP addresses. The program's details are described in Section 3.2.

We explore the throughput, RTT, and router CPU metrics

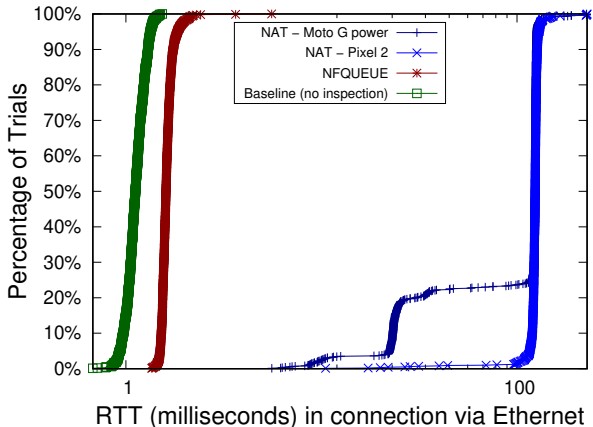

Figure 6: RTT with a log scale in milliseconds between the client and the server when the client connects to the router via Ethernet. The leftmost green line shows baseline result. The middle red line shows the result after applying on-router inspection via the NFQUEUE library. The two rightmost blue lines show the results with two separate phones after applying on-phone inspection using NAT redirection rules.

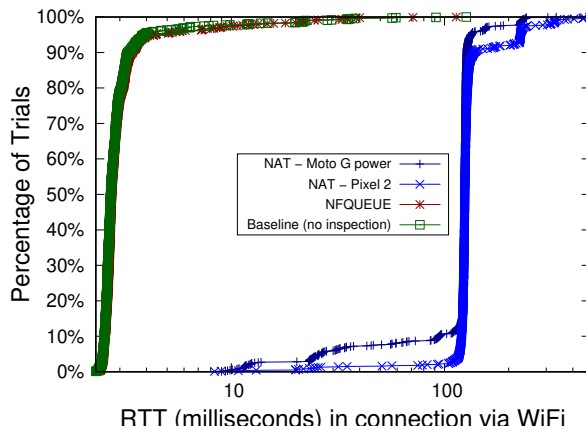

Figure 7: The RTT with a log scale in milliseconds between the client and the server when the client connects to the router via WiFi. The green line shows the baseline result. The red line shows the result after applying on-router inspection via NFQUEUE library. The two blue lines show the results with two separate phones after applying on-phone inspection using NAT redirection rules.

of the on-device inspection program using the same tools and settings used in Section 5.1. In the two left-most lines of Figures 3 and 4, we see the upload and download throughput after applying this inspection approach. We conducted 3 trials and measured the throughput for 1000 seconds after an ini-

tial delay of 100 seconds to accommodate TCP's slow-start behavior. As we seen in the right-most two lines in Figure 3, the median download throughput is 9.62 Mbps, and the median upload throughput is 8.40 Mbps (standard deviation of 3.91 Mbps for download and 3.93 Mbps for upload). Given this substantially decreased throughput from the baseline, we hypothesize that the change introduces a bottleneck on the router.

When we examine the CPU usage of the router, we confirm that this resource is exhausted. In Figure 5, we see that the baseline CPU usage is around 9% when throughput is limited to 10 Mbps, but is 100% when the router performs packet inspection. The process elevates all traffic to the router's Linux user space environment, which requires significant computational resources on the router. Such routers tend to be manufactured with lower-end CPUs for economic reasons [26] and there appears to be little headroom for this additional operation. However, when the router is not overwhelmed, as in the simple echo server RTT tests, we see that the on-device router introduces minimal RTT increases over the baseline. These results are shown by the red line in Figure 6, which is close to the baseline results.

## 5.3 On-Phone Inspection via NAT Redirection

With the CPU limitations of residential routers, we explore the potential of re-routing packets via a smartphone to inspect traffic. As described in Section 3, we add three different NAT rules via `iptables` on the router to cause traffic to be sent via the phone. An example of traffic forwarding, after applying NAT rules, is shown in Figure 1.

The NAT rules cause traffic to be sent to a specific port on the smartphone. Our Java program runs on the phone, binds to the specified port, and receives packets. It performs simplistic packet inspection and then sends it back to the router on a specific port. The router uses its NAT rewriting rules to send it on to the server. When a reply from the server is received by the router, the traffic likewise traverses the phone for inspection before traveling to the client.

We use the same three metrics as in the baseline and on-router cases to explore the performance characteristics of this phone-based inspection approach. In addition, we consider the CPU usage of the phone application itself, since high usage may result in battery depletion on the phone and could prevent its practical deployment.

Using the same settings as in the two prior sections, we explore the throughput when traffic is directed through the Moto G Power smartphone. In the middle two lines of Figures 3, we see that the median download throughput is 94.80 Mbps and the median upload throughput is 70.10 Mbps, with tight distributions (standard deviation of 4.32 Mbps for download and 2.87 Mbps for upload). The throughput is substantially higher than the on-router inspection approach in both Figure 3 and Figure 4. In effect, the processing of the NAT rules

on the router may incur less computational overhead than the full process of inspecting the traffic. Since the router's CPU was the bottleneck in the on-device inspection scenario, this adjustment increases the amount of traffic the router can handle.

In Figure 5, we are able to confirm that the NAT-based approach yields significantly lower CPU utilization than on-device inspection when throughput is limited to 10 Mbps. The middle line in that graph shows that the NAT approach has a median of 24.0% CPU utilization with a standard deviation of 2.61%.

The insertion of another device on the network path through a loop will necessarily increase the packet's propagation delay and may be observable in the overall end-to-end RTT. This is apparent in Figure 6, with the RTT of the NAT approach represented by the two right-most lines. We see patterns where 20% of traffic has an RTT less than 30.44 ms while 75% of traffic has an RTT over 120.17 ms. This is significantly higher than either the baseline scenario or when on-device inspection occurs. Importantly, this experiment uses a simple echo server approach and does not tax the CPU of the router. The on-device scenario would incur greater RTT delays when the CPU is a bottleneck due to processing delay.

In Figure 8, we explore the cause of the RTT delay in greater detail. We host a simple TCP echo program in three different ways. The left-most line represents the scenario when the echo server runs on the server using the baseline scenario (i.e., the traffic traverses the router to the server, bypassing the phone). The middle line represents the case when the echo server runs in an application within an Android emulator running on a laptop. The two rightmost lines represent the echo server running on two separate physical smartphones: a Moto G Power and a Pixel 2. While the first two scenarios have fairly tight distributions with RTTs less than 10 ms, the echo server built on the Moto G Power has a latency around 20ms for most traffic. However, it has much longer delays for around 20% of traffic. Moreover, the echo server built on the Pixel 2 has a latency of less than 50ms for around half of the trials, but has delays over 200 ms for around half of the traffic. In essence, the simple echo server smartphone application sometimes incurs significant delay in sending or receiving traffic. While this occurs only around 20% of the time for the echo server, the proxy example would incur two instances of this behavior, causing more traffic to incur a delay.

The distinct RTT behaviors exhibited by the two physical phones, that are not present in the Android emulator, may indicate some outside effect due to phone-specific factors. These could include the use of power savings modes, in which applications are periodically suspended or queued to reduce energy consumption.

Our last metric explores the energy usage of the proxy application on the phone. We again use the Moto G Power smartphone as a proxy while maximizing throughput transmission from the client to the server. In this experiment, we

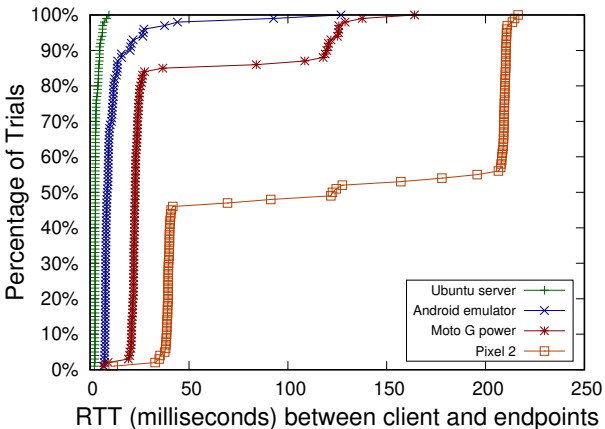

Figure 8: Comparison of RTT between connecting the client directly to the Ubuntu server, the Android emulator, the Moto G Power, and the Pixel 2.

also open a music-playing application on the phone, in the background, for comparison. We then record the CPU usage of the proxy application and the music application for 1000 seconds using the `top` tool in the phone. We monitor the idle percentage of the 8 cores in the proxy device. In Table 2, we show the CPU usage of the proxy application and the music application, along with the time for which the CPU core is idle. In this table, 100% represents the full utilization of a core on the device and 800% represents the full utilization of all eight device cores. The first row in Table 2 represents the proxy application, which uses only about 21% of a single core (and roughly 107% of the CPU usage of the music application). We see that the majority of the device's computational resources are unused. As a result, we anticipate that the CPU-based energy consumption of the device would be a small fraction of a music application. Since phones are regularly used for music playing without significant power-related disruptions to end-users, it is likely that the proxy application would likewise be accommodated by phones.

Table 2: CPU usage of the smartphone for different applications when maximizing throughput while applying on-phone inspection.

| Percentile of Trials | 10th | 50th | 90th |
|---|---|---|---|
| CPU Usage of Proxy App | 18% | 21% | 24% |
| CPU Usage of Music App | 98% | 107% | 114% |
| CPU Idle | 535% | 560% | 584% |

## 6 Conclusion

The need for privacy and the limited computational resources in residential networks complicate traffic inspection and anal-

ysis. Residential routers' limited CPU resources make it difficult to deploy even straightforward IP address-based inspection tools without substantially limiting throughput through the router. However, with carefully-crafted NAT rules, a router can redirect communication through another device, such as a smartphone, to inspect traffic.

In our experiments, we find that NAT-based diversion through a smartphone can substantially raise the communication throughput from around 10 Mbps to around 90 Mbps. The router can periodically examine its ARP and DHCP data structures to detect the availability of a phone in the LAN, contact an application on the phone to configure proxy services, and then divert traffic through the phone to enable outsourced inspection. With such an approach, residential networks can opportunistically use available smartphones as middleboxes to enable higher-throughput traffic inspection.

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
