# OpenReview forum: "Solution: Inspecting Traffic in Residential Networks with Opportunistically Outsourced Middleboxes"
_JSYS/2022/Aug_Papers — Reject_

### Official Review · Reviewer_62Uo · 2022-08-25
**The problem scope and the solutions are limited**

**Decision:**

Strong reject: this paper has serious problems, fixing it would definitely take more than three months

**Review:**

Thank you for submitting the paper to JSys.

## Summary of the paper
Home networks often lack security features that are available in enterprise networks due to the limited capabilities and resources in network hardware devices. This paper proposes a mechanism that enables further inspection of network traffic in home networks without enterprise-grade hardware equipment. The main idea is to re-route traffic to on-router CPUs or external computing devices (e.g., smartphones) connected to a router and process packets on the CPUs. The authors implement the mechanism to a hardware home router and a smartphone and evaluate the performance overhead.

## Strengths
Implementation and experiments with real hardware devices.

## Weaknesses
**[Unclear problem definition and limited problem scope]** I think the paper does not clearly specify the problem definition, and the scope of the problem is limited. In Section 2, the authors mention that prior work can suffer from high compute resource overhead due to additional on-device computation or potential privacy risks when offloading traffic to the cloud for further investigation. Given that, I assumed the goal of this paper would be to achieve the same goal as prior work (i.e., supporting advanced security features) while reducing the overhead or privacy risks. However, this paper seems to further limit the scope of the problem, which is supporting a simple blacklist-like functionality in a home network. In theory, they can run more advanced and complex functionality on the external CPUs, but I would expect high performance overhead (e.g., throughput degradation and latency increase) due to the limited compute power on the CPUs. I suggest the authors clarify what problem they try to solve and its scope first before describing prior work in the space.

**[Limited capabilities]** Unlike the prior work mentioned in Section 2, the mechanism described in the paper can support a limited set of functionality because it relies on compute/memory resources of routers or phones. The authors should clearly compare the proposed mechanism with the prior work in terms of security capabilities and performance/resource overhead. Also, it is unclear how the proposed mechanism ``opportunistically'' leverage external resources. The security features must always be on, which requires a smartphone to be always connected to the home network and dedicate a portion of its resource to packet inspection, which is not a practical deployment model.

**[Unclear technical challenges]** In Section 3, the paper explains how the authors implement on-router and on-phone inspection by inserting a few iptables rules that redirect traffic. Before jumping into the implementation of the solution, the authors should clarify key challenges in realizing their approach and key ideas to solve them.

**[Performance overhead]** The performance overhead due to traffic rerouting and inspection seems non-negligible. For example, Figure 4 shows that median uplink throughput reduces more than 2$\times$ (~229 Mbps vs. ~100Mbps) due to traffic redirection to the smartphone. The authors should explain why the overhead would be acceptable, especially compared to the prior work mentioned in Section 2.

In summary, I would suggest the authors clearly describe the requirements of a traffic inspection system for home networks, the design space of options, technical challenges in realizing the proposed approach, and key ideas.

**Expertise:**

Actively publishing in this area

**Useful:**

no

---

### Official Review · Reviewer_MjVN · 2022-09-02
**Review for Inspecting Traffic in Residential Networks with Opportunistically Outsourced Middleboxes**

**Decision:**

Weak reject: interesting papers with flaws, not sure if they can be fixed in three months

**Review:**

**Paper summary**

Home network security has become a concern given the increase of the smart home market. Existing home network security solutions have several limitations including 1) compute-intensive methods cannot run on home devices with limited resources and 2) third-party approaches raise privacy concerns. This paper proposes to inspect residential network traffic using available devices in the same network such as routers, smartphones, laptops, etc. Two approaches are discussed in the paper: 1) inspecting the network on the router, and 2) redirecting traffic to smartphones via NAT on the router. The authors verify these ideas by setting up a real residential network. The results show the feasibility of the proposal and the overhead incurred by the system.


**Strengths**

+ Good writing. Clear motivation and a good summary of existing approaches and their limitations.

+ Comprehensive performance evaluation of the proposed methods.


**Weaknesses**

- Lack of technical depth; only test with very simple traffic inspection

- Some design space is not considered, e.g., mobility of home devices, security of the middleboxes


**Comments**

Thanks for your submission. It was a nice reading; the paper is well-written. The background section helps me understand the state of the art and its limitations. The proposal of using home devices attached to the same residential network for traffic inspection makes sense to me. The experiments also verify the feasibility of the idea. However, I believe there is still a large space we can improve the paper.

- The evaluation is based on a very simple block-list-based approach, but the authors do not justify how practical the approach is. For example, how the block-list IPs are learned? If in practice more complicated solutions are needed, how much additional overhead will be incurred? If a block list already works well, why don't we insert these rules into the router directly rather than processing the traffic in the router's user space or redirecting it to a smartphone?

- For the on-router solution, sending traffic to the user space incurs significant overhead. Should we consider in-kernel packet processing using ebpf or xdp? In such a way, packets do not need to go into the user space, which could potentially reduce the overhead.

- For the on-phone inspection, the authors do not consider the mobility of the devices. Concretely, what if the phone moves out of the residential network? What should the system do to continue the traffic inspection? One possibility is to have a device manager on the router or somewhere else, which monitors the state of the devices in the same network. Then, there could be a scheduler that redirects traffic to different devices according to their availability.

- Because of phone mobility, it is possible that a phone is hacked outside the home network. Later, when the phone is attached to the home network, the attacker can see all traffic inspect on it. It would be good to discuss potential vulnerabilities and solutions here.

- The paper evaluates the energy usage of the on-phone inspection by comparing CPU cycles with a music app. This can indicate the power consumption, but a more direct metric is the battery statistics. For example, we can charge the phone to 100% battery, run the app for an hour, and observe the remaining percentage of the battery.

- Some minor writing issues:
   - In intro, "We use open source firmware on a consumer-grade router to profile traffic locally and via a smartphone.", consider changing to something like "We use open source firmware to profile traffic locally on a consumer-grade router and remotely on a smartphone." This is more symmetric and might be easier to understand.
   - The last sentence in the first paragraph of 2.2, "via a connection initiated from within the network." typo "from within"?
   - First paragraph of 3.3, "the packets sent from server will will also go in the reverse direction", typo "will will"
   - Second paragraph of section 4, "For the network configuration, as showen in Figure 2," typo "showen"

**Expertise:**

Follow the literature closely, last published 5+ years ago

**Useful:**

yes

---

### Official Review · Reviewer_Hady · 2022-09-09
**Review for paper on Inspecting Traffic in Residential Networks with Opportunistically Outsourced Middleboxes**

**Decision:**

Weak reject: interesting papers with flaws, not sure if they can be fixed in three months

**Review:**

## Paper summary
The paper evaluates the use of smartphone devices as a proxy to capture and analyse traffic between the client and the Internet. The smartphone is connected as a proxy to the router through NAT, traffic is redirected from the router to the smartphone, before being delivered to the server. It then compares the performance (throughput, latency and CPU usage) between NFQUEUE, smartphone and baseline measurements.

## Strengths
* The findings are interesting as they show that offloading the traffic inspection off the router has some performance gain.
* Concepts in the paper are explained fairly well.

## Weaknesses
* The number of trials (3) is too few for the statistics to be significant
* Paper investigated only one type of router and two smartphones, this is relatively too low
* No details about how results are affected by connectivity medium, congestion on the link, etc
* Proof-reading required

## Comments to the authors

Thank you for submitting this paper to JSys. The findings of the paper are interesting even if the results are pretty much expected, it is good to see some empirical analysis. However, I do find some major concerns with the way the experiment was conducted.

### Number/type of devices
First, the experiment was run only on a limited set of devices. Only two smartphones were tested. Authors should have rather tested multiple devices categorized as low-end, mid-end, and high-end devices. Similarly, multiple combinations of router software/hardware should have been tested. This would provide an idea of the performance spectrum.

### Connection medium
It seems the throughput test has been conducted using WiFi radio links only. It may be possible that a smartphone is connected via USB cable to the router using the router's USB port. If not possible, this should be stated in the document.

### Congestion
The performance can also vary based on how congested the traffic is and on the number of users. The congestion and other features such as packet loss, and jitter could help build the performance model. And even simulations could have been performed (probably in a future study). The paper does not talk about common performance degrading issues such as the famous "bufferbloat", which in this type of experiment, might not be negligible.

### Threat Model
I didn't quite catch the threat model in this paper, why is there a need for one? Here the idea is only to showcase the performance gain in using a specific technique for traffic inspection. How such a system is secured in the wild, in my opinion, is out of the scope of this study.

### Writing
The paper should be proofread, it has a lot of grammatical mistakes and some phrases are not well constructed. I'd advise to authors to get it fixed by an English speaker. E.g. "were worry", "as showen", "can peeking"

The abstract is too long and first paragraph is unnecessary, stick to the research questions and findings.

### Missing reference
In the introduction, the claim "In 2021, 442 million smart devices,and 82% home networks are connected to the Internet" is missing a reference.







**Expertise:**

Follow the literature closely, last published 5+ years ago

**Useful:**

yes

---

### Official Review · Reviewer_s1iY · 2022-09-13
**Interesting idea but not well-motivated or evaluated**

**Decision:**

Weak reject: interesting papers with flaws, not sure if they can be fixed in three months

**Review:**

Thank you for submitting this work to JSys. I enjoyed reading this paper.

That said, I got stuck with the motivation for the proposed work.

First, the paper needs to set up the problem statement. It should clarify what specific analysis (minimum) we need to perform on the network packets, the expected workload, acceptable latency overheads, etc. For example, I'd argue that most traffic from user devices, such as laptops and smartphones, will already be encrypted, and we don't need any further inspection. Do we expect a 100+ Mbps workload from smart devices? I don't know the answer, but I think understanding the workload is critical while designing such a system.

Second, the paper needs to argue better why cloud-based solutions are not reasonable here. Specifically, the solution where you can use a VPN service to tunnel your traffic to a nearby compute node and use the elastic cloud resources to perform the desired network functions. The paper argued that such a service would entail additional compute/service costs for the end user. True, but it is not clear that this cost is less than the proposed one where the end user will be using the CPU cycles of their smartphones (or other similar devices).

IMHO the latency overhead of routing the traffic to a nearby edge node is concerning. Still, those latency overheads are much better than the ones incurred by the proposed approach that can shoot up to 100+ ms.

I buy the argument that offloading NFs to the cloud raises privacy concerns. In that case, I'd argue why not consider a dedicated compute node for analyzing the network traffic. Single-board computers (e.g., RasPis) are cheap ($30-100) and can easily handle a few tens of Mbps workload w/o incurring any significant latency overheads (I am speculating here). My concern is that a smartphone/laptop will have much more going on beyond a simple music app, and the latency can fluctuate significantly, which is highly undesirable.

The paper does a great job developing a prototype. I can see myself using this setup to teach students how to use NFQUEUE and SNAT. It will be great if the authors make this research artifact publicly available.

That said, I have some concerns about the evaluation too.

First, it is hard to follow, and I suggest you add a subsection that describes the setup, performance metrics, and related methodology. You can then have the following two subsections on throughput and latency. That way, readers won't have to refer to a figure across different subsections.

Second, I think the paper selects a lightweight inspection task for evaluation. This choice makes it difficult to understand the cost of using the proposed design for a more realistic workload. It would be great if the paper quantifies the overheads for tasks discussed/motivated in the paper, e.g., IDS or some anomaly detection algorithm (references 8-10). I am concerned that performing these more intense tasks is too taxing for a smartphone and will show that the proposed design is impractical.

Finally, I think the paper should also report latency under load. I believe the latency inflation will be pretty significant for 100+ Mbps workloads.

Overall, I think the proposed design is intriguing, but it is not well motivated or evaluated. The current evaluation does not make a strong case for the proposed design.

**Expertise:**

Actively publishing in this area

**Useful:**

yes

---

### Meta-Review · Area_Chair_StTb · 2022-09-16

**Recommendation:** Reject
**Confidence:** 5

**Metareview:**

Thanks for submitting your paper to JSys! The reviewers appreciate the prototype development and performance evaluation on real hardware devices. However, all reviewers agree that the paper is not ready for publication in its current form due to a lack of clarity on the following fronts:
- Problem is not well-defined or motivated, e.g., requirements for the targeted traffic inspection system in home networks need to be articulated;
- Some straightforward alternative solutions in the design space are not considered;
- Performance overhead of traffic rerouting and inspection is not systematically evaluated;
- Technical challenges and the key insights for addressing them are unclear;
- Writing needs improvement, e.g., evaluation setup, metrics, and methodology are missing.

Hence, the paper is recommended for rejection. We hope that the detailed reviews will prove helpful in strengthening the paper in the future.

---

### Decision · Program_Chairs · 2022-09-17

Reject